# Depletion of Ly6G-Expressing Neutrophilic Cells Leads to Altered Peripheral T-Cell Homeostasis and Thymic Development in Neonatal Mice

**DOI:** 10.3390/ijms24097763

**Published:** 2023-04-24

**Authors:** Jessica Rühle, Marco Ginzel, Stefanie Dietz, Julian Schwarz, Trim Lajqi, Sandra Beer-Hammer, Christian F. Poets, Christian Gille, Natascha Köstlin-Gille

**Affiliations:** 1Department of Neonatology, University of Tübingen, D-72076 Tübingen, Germany; jessica.ruehle@med.uni-tuebingen.de (J.R.); stefanie.dietz@med.uni-tuebingen.de (S.D.); julian.schwarz@med.uni-tuebingen.de (J.S.); christian-f.poets@med.uni-tuebingen.de (C.F.P.); 2Department of Pediatric and Adolescent Surgery, Paracelsus Medical University Hospital, A-5020 Salzburg, Austria; m.ginzel@salk.at; 3Department of Neonatology, Heidelberg University Children’s Hospital, D-69120 Heidelberg, Germany; trim.lajqi@med.uni-heidelberg.de (T.L.); christian.gille@med.uni-heidelberg.de (C.G.); 4Department of Pharmacology, Experimental Therapy and Toxicology, Institute of Experimental and Clinical Pharmacology and Pharmacogenomic and ICePhA, University of Tübingen, D-72074 Tübingen, Germany; sandra.beer-hammer@uni-tuebingen.de

**Keywords:** neutrophils, MDSC, T-cells, thymus, T-cell development

## Abstract

Newborns and especially preterm infants are much more susceptible to infections than adults. Due to immature adaptive immunity, especially innate immune cells play an important role in a newborn’s infection defense. Neonatal neutrophils exhibit profound differences in their functionality compared to neutrophils of adults. In particular, neonates possess a relevant population of suppressive neutrophils, which not only inhibit but also specifically modulate the function of T-cells. In this study, we investigated whether neonatal neutrophils are already involved in T-cell development in the thymus. For this purpose, we used a newly developed model of antibody-mediated immune cell depletion in which we administered a depleting antibody to pregnant and then lactating dams. Using this method, we were able to sufficiently deplete Ly6G-positive neutrophils in offspring. We demonstrated that the depletion of neutrophils in newborn mice resulted in altered peripheral T-cell homeostasis with a decreased CD4+/CD8+ T-cell ratio and decreased expression of CD62L. Neutrophil depletion even affected T-cell development in the thymus, with increased double positive thymocytes and a decreased CD4+/CD8+ single positive thymocyte ratio. Altogether, we demonstrated a previously unknown mechanism mediating neutrophils’ immunomodulatory effects in newborns.

## 1. Introduction

Sepsis is a major cause of death in the neonatal period. About 0.05% of term-born neonates suffer from sepsis, with its incidence rising to 36% in extremely preterm infants [1,2]. Due to highly limited antigen exposure of the fetus in utero, neonatal adaptive immunity is naïve, and newborns must rely on innate immunity to protect from early infections [3].

Neutrophils are the first-line circulating effector immune cells responding to and defending from infections. However, there are significant differences in neutrophil immune responses between adults and neonates. In addition to a markedly reduced storage pool [4], neonatal neutrophils, especially preterm infant neutrophils, exhibit various functional deficits compared to those in adults, such as reduced chemotaxis, adhesion, and transmigration [3,5,6], as well as diminished production of antimicrobial proteins [7] and reduced phagocytosis [8]. Furthermore, newborns have high numbers of neutrophil cells with immune-suppressive characteristics, which are the so-called granulocytic/polymorphonuclear myeloid-derived suppressor cells (GR-MDSC or PMN-MDSC) [9,10,11]. GR-MDSC expand under various pathological conditions, usually leading to harmful immunosuppression, especially by targeting T-cells [12]. In recent years, however, it has become more and more clear that GR-MDSC also accumulate physiologically during pregnancy in the maternal and fetal organism [9,13], facilitating maternal-fetal tolerance [14,15,16] and modulating neonatal immune responses [17,18,19]. GR-MDSC carry the same surface markers as neutrophils and can be distinguished from them only by their suppressive capacity and—in humans—by their lower density [20,21]. Thus, the neutrophils of newborns represent a heterogenous mixture of proinflammatory and suppressive acting immune cells.

A previous study in newborn mice showed that the totality of neutrophilic cells provided protection from sepsis [22], but it remained unclear whether this protective effect was mediated by the pro- or anti-inflammatory properties of neutrophilic cells. Later on, another study specifically showed that GR-MDSC provided protection from necrotizing enterocolitis (NEC) [23]. This protective effect was through T-cell suppression [23]. GR-MDSC use various effector mechanisms, with the most prominent being arginase 1 (ARG-1), nitric oxide (NO), and an upregulation of reactive oxygen species (ROS) [12]. Via these mechanisms, GR-MDSC of newborns not only cause a general suppression of T-cells but can also specifically modulate their functions and, e.g., trigger the polarization of T-helper (Th) cells towards Th2 and induce the activation of regulatory T-cells (Tregs) [9,17]. Whether the T-cell homeostasis of newborns is influenced by neutrophils already during T-cell development in the thymus is yet unclear.

In the present study, we used a newly established model of depletion of Ly6G-expressing neutrophils (including classical neutrophils and GR-MDSC) in newborns through antibody application to the pregnant and then lactating dam. In this model, we showed that depletion of neutrophils in newborn mice leads to altered peripheral T-cell homeostasis, decreased expression of CD62L on T-cells, and alterations in thymic T-cell development. Thus, we demonstrated a new mechanism of how neutrophils can influence the immune responses at the beginning of life.

## 2. Results

### 2.1. Application of Anti-Ly6G Antibody to Dams Sufficiently Depletes Neutrophils in the Offspring

First, we developed a new model for the antibody-mediated depletion of Ly6G-expressing neutrophils in neonatal mice. We administered an anti-Ly6G antibody to pregnant dams beginning two days before the estimated date of delivery. Antibody application was repeated every three days to suckling mothers until pups were sacrificed. This resulted in a reduction in the percentage of Ly6G-positive cells in the pups’ spleens, livers, and lungs (other organs were not investigated) from a median of 8.8% to 0.8% at P7 and from median 2.0% to 0.3% at P14 (*p* < 0.0001, Figure 1A,D) in the spleen. Neutrophilic cells in livers and lungs decreased from a median of 20.7% to 2.7% (liver) and a median of 13.1% to 1.9% (lung) at P7 and from a median of 7.1% to 1.8% (liver) and a median of 10.4% to 1.7% (lung) at P14 (all *p* < 0.0001, Figure 1B,C,E,F). Thus, we demonstrated that the successful depletion of Ly6G-expressing cells via the mother is possible in newborn mice.

### 2.2. Depletion of Neutrophils Leads to Altered T-Cell Homeostasis

We have shown in previous work that the totality of neutrophilic cells in newborn mice exhibits suppressive properties, suggesting that neonatal neutrophils are, to a large percentage, GR-MDSC. Since the main target cells of GR-MDSC are T-cells, we next analyzed the T-cell composition in the spleens, livers, and lungs of the pups with and without Ly6G depletion. We found slightly increased levels of total T-cells in the spleens of anti-Ly6G treated pups compared to control animals at P7 (median of 34.4% vs. 28.8%, *p* < 0.01, Figure 2A) and in livers at P14 (median of 29.0% vs. 26.6%, *p* < 0.05, Figure 2E), while no differences in total CD3+ levels were observed in lungs upon Ly6G depletion (Figure 2I). Furthermore, we found decreased CD4^+^ to CD8^+^ T-cell ratios in all organs investigated at P7 after Ly6G depletion (2.0 ± 0.2% vs. 2.8 ± 0.5%, *p* < 0.05 for spleen, median 0.4% vs. 0.8%, *p* < 0.01 for liver, 2.0 ±1.2% vs. 3.9 ± 0.6%, *p* < 0.01 for lung) and in spleens also at P14 (2.2 ± 0.1% vs. 2.6 ± 0.2%, *p* < 0.01), while CD4^+^ to CD8^+^ T-cell ratio in lungs was increased after Ly6G depletion at P14 (3.6 ± 0.8% vs 2.1 ± 0.2%, *p* < 0.01) (Figure 2D,H,L). Figure 2B,C,F,G,J,K show percentages of CD4^+^ and CD8^+^ T-cells in pups with and without Ly6G depletion at P7 and P14, while Appendix A shows the gating strategy for total T-cells and CD4^+^ and CD8+ T-cells. In summary, we demonstrated that neutrophil depletion in newborn mice alters CD4^+^/CD8^+^ T-cell homeostasis.

### 2.3. Depletion of Neutrophils Leads to Decreased CD62L Expression on T-Cells

Next, we investigated T-cell activation status in pups with or without Ly6G depletion by analyzing CD62L expression on T-cells from spleens, livers, and lungs. We found a markedly decreased expression of CD62L on splenic and lung CD4^+^ (MFI 3129 ± 441 vs. 6240 ± 911, *p* < 0.0001 for spleen and 3650 ± 218 vs. 5940 ± 967, *p* < 0.01 for lung) and CD8^+^ T-cells (3028 ± 476 vs. 8066 ± 1402, *p* < 0.0001 for spleen and 5169 ± 304 vs. 8262 ± 2592, *p* < 0.01 for lung) after Ly6G depletion at P7 (Figure 3A–C,G–I). CD62L expression on liver T-cells was also slightly but not statistically significantly decreased on CD4^+^ T-cells (2036 ± 264 vs. 2993 ± 982, *p* = 0.07) at P7 after Ly6G depletion (Figure 3D–F). The decreased expression of CD62L on T-cells after depletion of neutrophilic cells in newborn mice may indicate enhanced T-cell activation.

### 2.4. Depletion of Neutrophils Leads to Altered Thymic T-Cell Development

Lastly, we asked whether Ly6G depletion in neonatal mice may affect T-cell development in the thymus. In the thymus, only very low numbers of myeloid cells and neutrophils were present overall. However, the administration of the Ly6G antibody also led to a sufficient reduction of neutrophil cells in the thymus (Appendix A). We analyzed T-cell precursors in the thymi of anti-Ly6G-treated pups and control pups. Thymocytes mature from double negative (DN) cells over double positive (DP) cells to single positive (SP) cells. We found increased levels of DP and decreased levels of SP thymocytes after Ly6G depletion at P14 (Figure 4A–D), leading to a decreased SP to DP ratio (median 0.1 vs. 0.5, *p* < 0.01, Figure 4E), while the DP to DN ratio was markedly increased (median 36.9% vs. 12.5%, *p* > 0.05, Figure 4F). Looking at the T-cell subpopulations, we found decreased levels of SP CD4^+^ thymocytes at P14 after Ly6G depletion (median 9.6% vs. 31.1%, *p* < 0.01, Figure 4G), while levels of SP CD8^+^ thymocytes were increased (median 2.2% vs. 1.4%, *p* < 0.01, Figure 4H), leading to a strongly decreased CD4^+^ to CD8^+^ thymocyte ratio (median 0.1 vs. 0.5, *p* < 0.01, Figure 4I). Percentages of CD69^−^/CD62L^+^ cells, which leave the thymus, were only marginally altered by Ly6G depletion (Figure 4J–L). Within the DN subsets, CD25^+^/CD44^−^ DN3 cells were especially increased, and CD25^+^/CD44^+^ DN2 cells decreased after Ly6G depletion at P14 (Appendix A). In conclusion, our results show that neutrophilic cells in the neonate appear to modulate T-cell development in the thymus.

## 3. Discussion

Despite the increasing research in this area, the role of neutrophils in neonatal immune regulation is still incompletely understood. Whereas neutrophils in the adult have the primary function of defending against infection by phagocytosis and the killing of bacteria, production of antimicrobial peptides, and production of cytokines and chemokines to activate other immune cells, the neutrophils of newborns have defects in these functions and even exhibit immunosuppressive and immunomodulatory properties, especially via influencing T-cell functions [6,9,17]. In this study, we developed a new model to deplete neutrophils in neonatal mice and investigated the effect of such depletion on T-cell homeostasis.

Here, we depleted neutrophils in newborn mice by administering an anti-Ly6G antibody every three days to pregnant and subsequently to lactating dams. The antibody was transferred into the circulation of the fetuses/offspring, leading to an efficient reduction of Ly6G-expressing cells. This transfer is only possible because of the expression of the so-called neonatal Fc-Receptor (FcRn). FcRn is expressed on the placenta in both human and mice, mediating a transplacental transport of IgG during pregnancy, but unlike in man, newborn mice also express FcRn on intestinal epithelial cells, which allows IgG antibodies to also be transported postpartum from the intestinal lumen into the circulation [24]. The standard procedure for antibody application to newborn mice is currently an IP injection [23,25,26,27]. For antibodies with a relatively short duration of action, this means frequent injections and stress for the experimental animals. In addition, repeated IP injections could lead to immune reactions in the peritoneal cavity, which are triggered by the injections themselves and not by the applied antibody. The method we have now established is an elegant way to circumvent these problems. As the FcRn is also expressed on the placenta [24], antibody application to pregnant dams may also be useful for administering antibodies to fetuses and thus avoiding costly procedures with surgery in the dams [28].

We found decreased levels of CD4^+^ T-cells and increased levels of CD8^+^ T-cells after the depletion of neutrophils in the spleens and livers of neonatal mice, leading to a decreased CD4^+^ to CD8^+^ T-cell ratio. It is known that newborns have higher CD4^+^ levels and an increased CD4^+^ to CD8^+^ ratio compared to adults [29]. Preterm infants also exhibited an increased CD4^+^ to CD8^+^ T-cell ratio compared to full-term neonates [30], suggesting that an elevated CD4^+^ to CD8^+^ ratio characterizes a more immature immune system. Under this assumption, our results would imply that neutrophils in neonates mediate a slowdown in the maturation of adaptive immunity. Considering that neutrophils are involved in the establishment of the microbiome in neonates [22,31], a delayed maturation of adaptive immunity might be useful to tolerate the colonization of body surfaces with microbes without overwhelming inflammation. However, these are pure speculations, and the role of neutrophil/T-cell interaction in this context needs to be investigated in further studies.

Interestingly, increased CD8^+^ T-cell numbers were found in the offspring of mice with maternal immune activation (MIA) [32]. The latter is associated with impaired neurological development [33]. The role of fetal/neonatal neutrophils in this context should also be evaluated in further studies. Looking at the lungs, we found increased CD8^+^ T-cell levels after neutrophil depletion at P7 but decreased levels at P14. Interestingly, a report investigating immune cell populations in the context of RSV infection in mice with intrauterine exposure to tobacco smoke found increased numbers of neutrophils accompanied by decreased numbers of CD8^+^ T-cells [34], in line with our results at P7. The cause and clinical significance of a reversal of this effect seen at P14 are unclear. Kusmartsev et al. previously reported that immature neutrophilic cells suppress CD8 T-cells via the production of reactive oxygen species [35]. In turn, ROS production is negatively regulated, among others, by the transcription factor hypoxia inducible factor 1α (HIF-1α) [36], which also plays an important role in the regulation of MDSC function [15,16,37]. We have recently shown that HIF-1α expression is substantially reduced in neonatal mononuclear cells compared to adult mononuclear cells [38], possibly due to adaptation of the neonatal immune system to hypoxia in utero. A lack of repression of ROS production by HIF-1α in the early neonatal period may cause neutrophilic cells of the newborn to produce high levels of ROS, thereby suppressing CD8^+^ T-cells in particular. Maturation with the increasing activability of HIF-1α and thus the suppression of ROS production would then explain that at P14, neutrophil depletion no longer has inducing effects on CD8^+^ T-cells. However, these considerations are pure speculation and should be verified in further studies.

Next, we found a decreased expression of CD62L on CD4^+^ T-cells in all organs investigated and on CD8^+^ T-cells in spleens and lungs at P7. CD62L is a selectin expressed on most leukocytes and is shed from the surface upon activation [39]. This indicates enhanced T-cell activation after neutrophil depletion and would fit with the suggestion that a large proportion of neutrophils in newborn mice are suppressively acting neutrophils/GR-MDSC. We recently observed a similar effect in pregnant mice in which a genetic alteration (absence of the MHC-I molecule Qa2) resulted in impaired GR-MDSC expansion during pregnancy, accompanied by decreased expression of CD62L on T-cells [16]. However, our results contrast with work by other groups, which showed that MDSC lead to a downregulation of CD62L on T-cells [14,40]. CD62L is also necessary for the homing of naïve T-cells to lymph nodes [39]. Unfortunately, we did not examine peripheral T-cell subtypes, so no conclusion can be drawn as to whether the ratio of naïve, memory, and effector T-cells was affected by neutrophil depletion.

Finally, we observed a profound change in T-cell development stages in the thymi of newborn mice with increased DP thymocytes and decreased DN and SP thymocytes. The population of DP thymocytes is by far the largest in the thymus. At this stage, positive selection (i.e., the survival only of cells capable of recognizing peptide: MHC complexes) occurs, leading to the death of 90% of all DP thymocytes. The large pool of DP thymocytes results from a massive expansion of DN thymocytes [41]. Since one of the most important abilities of neutrophils in newborns is the inhibition of T-cell proliferation [9,23], it can be speculated that the depletion of neutrophils abolishes these inhibitory properties and allows DN thymocytes to proliferate better, resulting in an increased pool of DP thymocytes. This is supported by the fact that within the DN thymocytes the stage of DN3 cells was increased after neutrophil depletion in which a particularly pronounced proliferation takes place [41]. An analysis of the CD4^+^ and CD8^+^ subpopulations of mature SP thymocytes revealed, as in the other organs, a decreased proportion of CD4^+^ T-cells after neutrophil depletion but an increased proportion of CD8^+^ T-cells compared with control animals.

At this point, an important limitation of our work must be mentioned. Unfortunately, we did not perform cell counting of thymocytes but only determined the percentages of subpopulations. For this reason, we cannot make any statement as to whether absolute numbers of thymic mature CD4^+^ or CD8^+^ T-cell numbers are changed by neutrophil depletion. However, thymus size was similar in depleted and non-depleted animals, suggesting that thymus development was not generally affected. Another limitation is that we analyzed only two to three litters per parameter. For individual parameters such as DN and DP thymocytes at P14, there was a clear clustering depending on the litter. This could possibly be explained by a more or less efficient depletion. Further studies on the direct influence of neutrophils on thymic T-cell development in newborns are necessary to better understand the immune cell interactions during this phase. A third important limitation is that we did not perform any functional neutrophil analyses in this study, which could have shed light on whether the neutrophils that were depleted in our model actually have suppressive properties. However, we know from previous experiments that CD11b^+^/Ly6G^+^ cells from the spleens of newborn mice at P7 sufficiently inhibit T-cell proliferation (own unpublished data), so we actually assume that neutrophils are suppressive rather than proinflammatory in newborn mice in toto.

In summary, with this work, we were able to describe a new and effective model for antibody-mediated immune cell depletion in newborn mice to show that neutrophils in the newborn not only modulate T-cell properties in the periphery [17] but are already involved in T-cell development in the thymus. We thus show a new aspect of the role of neutrophils in newborns. More detailed investigations of the underlying mechanisms are necessary to develop new approaches to possibly influence immune maturation in newborns favorably.

## 4. Materials and Methods

### 4.1. Mice

C57BL/6J mice were obtained from Charles River (Sulzfeld, Bavaria, Germany). Animals were maintained under pathogen-free conditions in the research animal facility of Tuebingen University, Tuebingen, Germany (K08/19G). All experimental animal procedures were conducted according to the guidelines from Directive 2010/63/EU of the European Parliament on the protection of animals used for scientific purposes and the German federal and state regulations.

Neutrophils were systemically depleted in newborn mice by administering a depleting Ly6G antibody to pregnant and subsequently to lactating dams. For this purpose, wild-type mice (C57BL/6J) were terminally mated. Two days before the expected litter date, the dam received an intravenous (IV) injection of a depleting anti-Ly6G antibody (clone 1A8, BioXCell, Lebanon, NH, USA) or an isotype control antibody (IgG2A, BioXCell, Lebanon, NH, USA) into the tail vein. This achieved depletion of all peripheral neutrophils. Because reconstitution of cells from the bone marrow occurs within three days [42], the injection was repeated every three days until the offspring were sacrificed. The first injection after birth was again given IV into the tail vein of the dam, and all subsequent injections were given intraperitoneally (IP).

### 4.2. Tissue Collection and Single Cell Preparations

Newborn mice were sacrificed on postnatal days 7 (P7) and 14 (P14), and spleens, lungs, livers, and thymi were removed. Tissue was homogenized using a 40µm filter (Greiner bio-one, Frickenhausen, Baden-Württemberg, Germany) and a syringe plunger to obtain single cell suspensions. All single cell suspensions were then adjusted to 1–5 × 10^6^ cells/mL in PBS.

### 4.3. Flow Cytometry

For extracellular staining of mouse cells, freshly isolated cells were washed in FACS buffer and fluorescent-conjugated extracellular antibodies added. Antibodies were purchased from BD Biosciences, Heidelberg, Baden-Württemberg, Germany (CD3 FITC (145-2C11; cat.# 553062), CD4 APC (RM4-5; cat.# 561091), CD4 FITC (RM4-5; cat.# 553046), CD8a APC-H7 (53-6.7cat.# 560182), CD11b FITC (M1/70; cat.# 553310), CD19 PE (1D3; cat.# 557399), CD44 BB700 (IM7; cat.# 566506), CD45 PerCp (30-F11; cat.# 550994), CD62L BV421 (MEL-14; cat.# 562910), FSV510 Amcyan; cat.# 564406) and from BioLegend, San Diego, California (CD4 APC-Cy7 (GK1.5; cat.# 100414), CD11b PE (M1/70; cat.# 101207), CD25 APC (PC61; cat.# 102012), CD69 PE-Cy7 (H1.2F3; cat.# 104512), CD44 BV421 (IM7; cat.# 103040), Ly-6G APC (1A8; cat.# 127614), NK1.1 PE (PK136; cat.# 108707), Gr-1 PE (RB6-8C5; cat.# 108407), TER-119 PE (TER-119; cat.# 116207)).

For immune cell quantification in spleens, livers and lungs, cells were pre-gated to living CD45^+^ cells. Among these, immune cell subsets were identified as follows: T-cells CD3^+^, T-Helper cells CD3^+^/CD4^+^, cytotoxic T-cells CD3^+^/CD8^+^, and neutrophils CD11b^+^/Ly6G^+^. For thymocyte quantification, cells were pre-gated to lineage negative cells (lineage cocktail consisting of CD19, CD11b, NK1.1, Gr-1, TER-119). DN cells were identified as CD4^−^/CD8^−^ cells, DP cells as CD4^+^/CD8^+^, and SP cells as CD4^+^/CD8^−^ or CD4^−^/CD8^+^ cells. Among DN thymocytes, development stages were differentiated by staining with CD25 and CD44. Among SP thymocytes, development stages were differentiated by staining with CD62L and CD69. Appendix A shows gating strategies for splenocytes and thymocytes. Data acquisition was performed with a FACS Canto II flow cytometer (BD Bioscience, Haryana, India) and analyzed via FlowJo V10 (FlowJo, LLC, Ashland, OR, USA).

### 4.4. Statistical Analysis

Statistical analysis was done using GraphPad Prism 9.1.2 (GraphPad Software, La Jolla, CA, USA). Comparisons between two groups of unpaired and not normally distributed data were evaluated using the Mann–Whitney test. A *p*-value < 0.05 was considered statistically significant. 

## Figures and Tables

**Figure 1 ijms-24-07763-f001:**
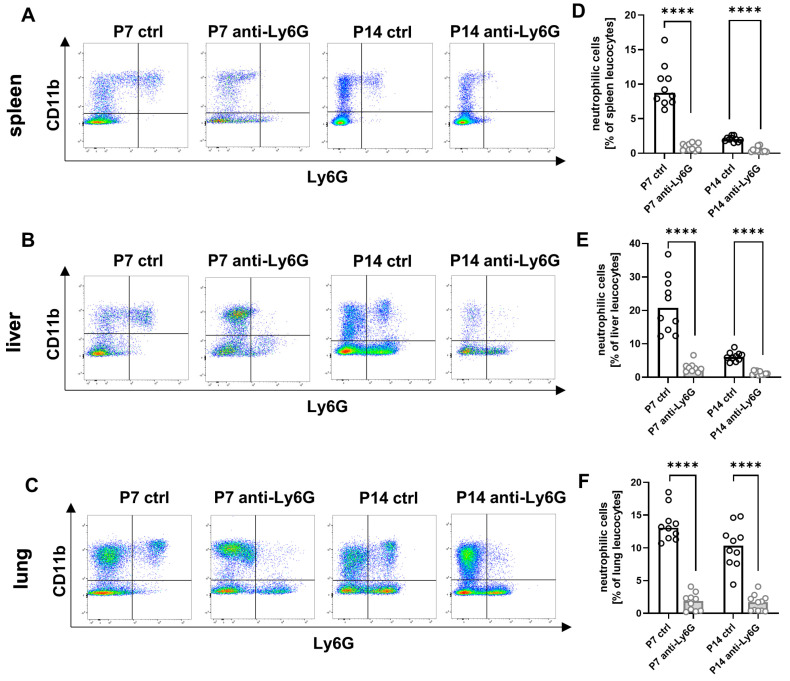
Depletion of neonatal neutrophils by antibody application to the dam. C57BL/6J mice were terminally mated. Two days before the expected litter date, the dam received an intravenous (i.v.) injection of a depleting anti-Ly6G antibody or an isotype control antibody into the tail vein. Injections were repeated every three days until the offspring were sacrificed. Offspring were euthanized on the seventh postnatal day (P7) and the 14th postnatal day (P14), and organs were collected. Tissue was homogenized and filtered to obtain single cell suspensions. Cells were then analyzed by flow cytometry. (**A**–**C**) Density plots showing the population of Ly6G^+^/CD11b^+^ neutrophils in spleens (**A**), livers (**B**), and lungs (**C**) of newborn WT newborn mice without (ctrl) or with (anti-Ly6G) neutrophil depletion. Cells were pre-gated on living CD45^+^ cells. (**D**–**F**) Scatter diagrams with bars showing percentages of neutrophils from CD45^+^ leucocytes in spleens (**D**), livers (**E**), and lungs (**F**) of newborn mice without (white bars) and with (gray bars) neutrophil depletion at P7 and P14. Each circle represents an individual sample, and the median is indicated; *n* = 9–11, **** *p* < 0.0001; Mann–Whitney test.

**Figure 2 ijms-24-07763-f002:**
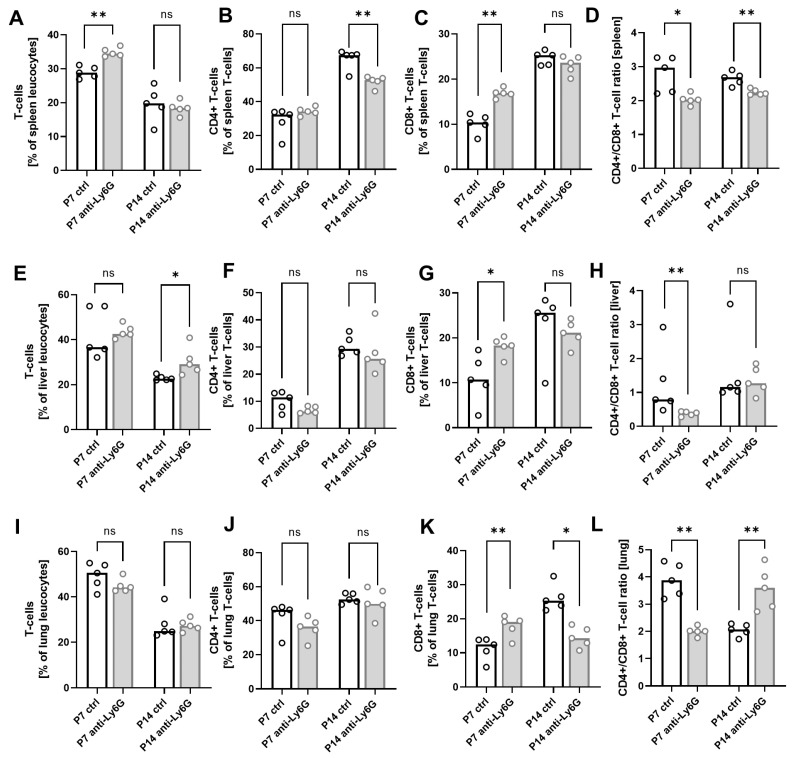
Peripheral T-cell subsets after neutrophil depletion in neonatal mice. C57BL/6J mice were terminally mated. Two days before the expected litter date, the dam received an intravenous (i.v.) injection of a depleting anti-Ly6G antibody or an isotype control antibody into the tail vein. Injections were repeated every three days until the offspring were sacrificed. Offspring were euthanized on the seventh postnatal day (P7) and the 14th postnatal day (P14), and organs were collected. Tissue was homogenized and filtered to obtain single cell suspensions. Cells were then analyzed by flow cytometry. (**A**–**L**) Scatter diagrams with bars showing percentages of total T-cells (**A**,**E**,**I**), CD4^+^ T-cells (**B**,**F**,**J**) and CD8^+^ T-cells (**C**,**G**,**K**) from CD45^+^ leucocytes as well as CD4^+^/CD8^+^ T-cell ratio (**D**,**H**,**L**) in spleens (**A**–**D**), livers (**E**–**H**), and lungs (**I**–**L**) of newborn mice without (white bars) and with (gray bars) neutrophil depletion at P7 and P14. Each circle represents an individual sample, and the median is indicated; *n* = 5, * *p* < 0.05; ** *p* < 0.01; ns not significant; Mann–Whitney test.

**Figure 3 ijms-24-07763-f003:**
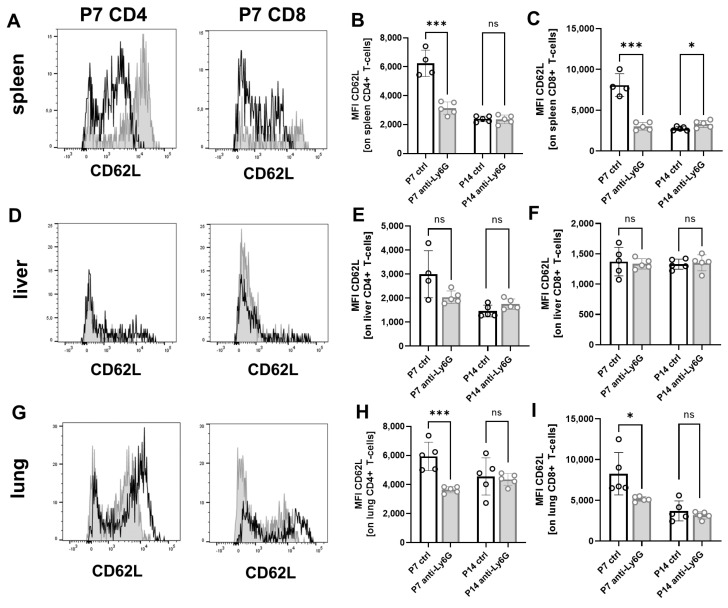
Peripheral T-cell activation after neutrophil depletion in neonatal mice. C57BL/6J mice were terminally mated. Two days before the expected litter date, the dam received an intravenous (i.v.) injection of a depleting anti-Ly6G antibody or an isotype control antibody into the tail vein. Injections were repeated every three days until the offspring were sacrificed. Offspring were euthanized on the seventh postnatal day (P7) and the 14th postnatal day (P14), and organs were collected. Tissue was homogenized and filtered to obtain single cell suspensions. Cells were then analyzed by flow cytometry. (**A**,**D**,**G**) Histograms showing expression of CD62L on CD4^+^ T-cells (left panel) and CD8^+^ T-cells (right panel) from spleens (**A**), livers (**D**), and lungs (**H**) of newborn mice without (empty histograms) and with (grey filled histograms) neutrophil depletion at P7. (**B**,**C**,**E**,**F**,**H**,**I**) Scatter diagrams with bars showing mean fluorescence intensity for expression of CD62L on CD4^+^ T-cells (**B**,**E**,**H**) and CD8^+^ T-cells (**C**,**H**,**I**) in spleens (**B**,**C**), livers (**E**,**F**) and lungs (**H**,**I**) of newborn mice without (white bars) and with (gray bars) neutrophil depletion at P7 and P14. Each circle represents an individual sample, and mean and standard deviation are indicated; *n* = 5, * *p* < 0.05; *** *p* < 0.001; ns not significant; unpaired *t*-test.

**Figure 4 ijms-24-07763-f004:**
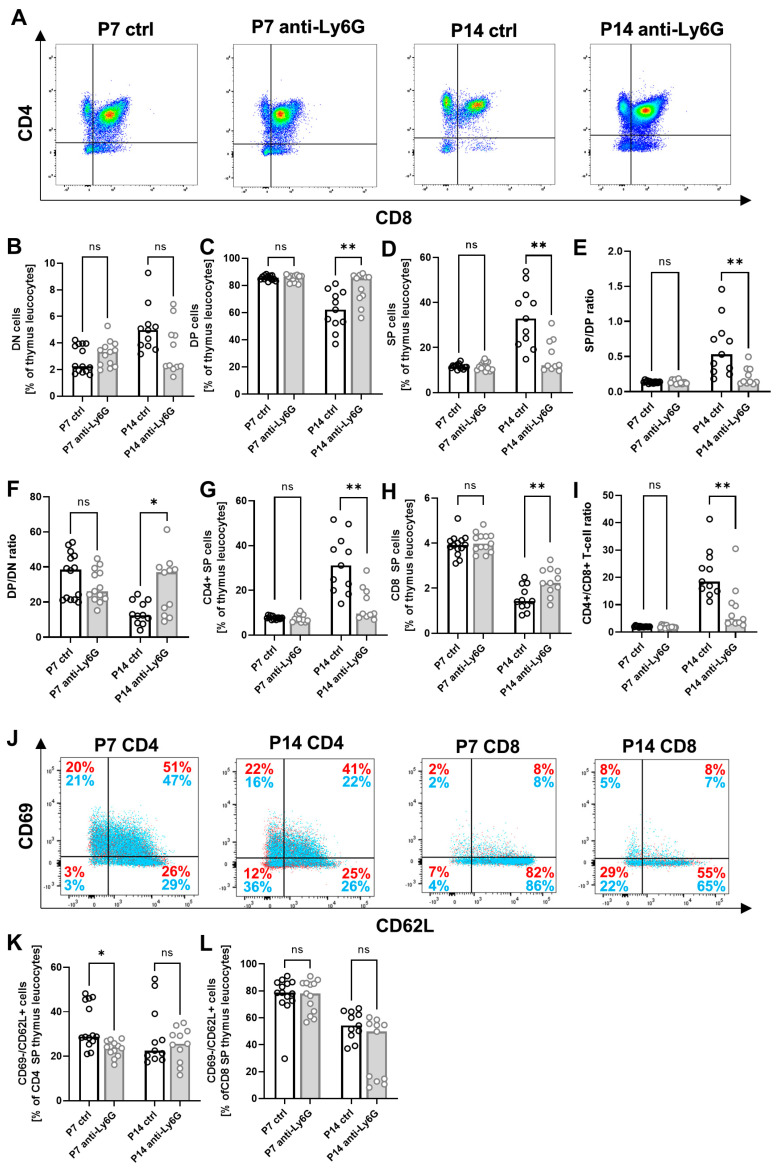
Thymic T-cell maturation after neutrophil depletion in neonatal mice. C57BL/6J mice were terminally mated. Two days before the expected litter date, the dam received an intravenous (i.v.) injection of a depleting anti-Ly6G antibody or an isotype control antibody into the tail vein. Injections were repeated every three days until the offspring were sacrificed. Offspring were euthanized on the seventh postnatal day (P7) and the 14th postnatal day (P14), and thymi were collected. Tissue was homogenized and filtered to obtain single cell suspensions. Cells were then analyzed by flow cytometry. (**A**) Density plots for CD8 vs. CD4 showing the populations of single positive (SP), double negative (DN) and double positive (DP) thymocytes in thymi of newborn WT mice without (ctrl) or with (anti-Ly6G) neutrophil depletion at P7 (left) and P14 (right). Cells were pre-gated on living lineage negative cells. (**B**–**I**) Scatter diagrams with bars showing percentages of DN (**B**), DP (**C**) and SP (**D**) thymocytes, SP/DP (**E**) and DP/DN (**F**) cell ratio, SP CD4+ (**G**) and SP CD8+ thymocytes (**H**) and CD4+/CD8+ cell ratio (**I**) in thymi of newborn mice without (white bars) and with (grey bars) neutrophil depletion at P7 and P14. (**J**) Density plots for CD62L vs. CD69 showing the populations of CD62L+/Cd69- thymocytes in thymi of newborn WT mice without (red) or with (blue) neutrophil depletion at P7 and P14. Cells were pre-gated on living lineage negative cells and CD4 SP or CD8 SP cells. (**K**,**L**) Scatter diagrams with bars showing percentages of CD62L+/CD69 CD4 SP (**K**) and CD8 SP (**L**) thymocytes in thymi of newborn mice without (white bars) and with (grey bars) neutrophil depletion at P7 and P14. Each circle represents an individual sample, and the median is indicated; *n* = 11–14, * *p* < 0.05; ** *p* < 0.01; ns not significant; Mann–Whitney test.

## Data Availability

The data presented in this study are available on request from the corresponding author upon reasonable request.

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
