# Peer review of "Depletion of Ly6G-Expressing Neutrophilic Cells Leads to Altered Peripheral T-Cell Homeostasis and Thymic Development in Neonatal Mice"

_ijms, 2023, doi:10.3390/ijms24097763_

Round 1
Reviewer 1 Report
The authors present a novel model of IgG-mediated neutrophil depletion and show that this depletion of neutrophils in the mothers pre- and post-partum leads to noticeable changes in the neutrophil and T cell populations in the neonates.
Minor changes:
-
Line 38: The word “important” doesn’t fit well in the given context. Please rephrase to “major” or something else.
-
Line 84: I believe the authors mean “pups” instead of “pubs”. This should be corrected in other places in the manuscript as well.
-
Line 85: Oxford comma usage is encouraged for scientific writing. This is missing in several other places in the manuscript as well.
-
Line 87-90: Please report p-values wherever differences in median/mean values of the dependent variable are discussed in the text.
-
Figure 1: Please provide error bars depicting S.D. wherever bar graphs are presented throughout the manuscript.
-
Line 128: The conclusion of sub-section 2.3 is too strong for the data presented. To support the conclusion that depletion of neutrophils leads to increased T cell activation, the experiment needs to be repeated with more activation markers assessed in vivo and/or ex vivo analysis of isolated T cells in context of proliferation/cytokine production. Otherwise, the conclusion of this section needs to be tempered.
-
Line 191: the word “applying” doesn’t seem appropriate. Please rephrase.
-
Line 194: It may provide more context to the readers if the authors include the fact that FcRn-mediated IgG transfer does take place between fetus and placenta in humans, similar to mice, but this doesn’t happen post-partum in human gut as it does in mice.
-
Line 284: Please provide a justification for using a different route of administration for the isotype control.
-
Line 306: Please also provide a supplemental figure depicting the gating scheme used.
-
Please add a summary or concluding statement at the end of each result sub-section.
Author Response
- The authors present a novel model of IgG-mediated neutrophil depletion and show that this depletion of neutrophils in the mothers pre- and post-partum leads to noticeable changes in the neutrophil and T cell populations in the neonates.
Thank you for your positive feedback on our manuscript. In the following, we will respond to your comments point by point.
- Line 38: The word “important” doesn’t fit well in the given context. Please rephrase to “major” or something else.
- Line 84: I believe the authors mean “pups” instead of “pubs”. This should be corrected in other places in the manuscript as well.
- Line 85: Oxford comma usage is encouraged for scientific writing. This is missing in several other places in the manuscript as well.
- Line 87-90: Please report p-values wherever differences in median/mean values of the dependent variable are discussed in the text.
Thank you for these comments. We addressed all the issues in the revised manuscript.
- Figure 1: Please provide error bars depicting S.D. wherever bar graphs are presented throughout the manuscript.
Thank you also for this point. On the bar graphs in Figures 1,2 and 4 medians are shown (as data are percentages and not assumed to be normally distributed), therefore we think it does not make sense to show standard deviations here. If you would still like us to do this, we are happy to revise i.t In Figure 3, we have followed your suggestion, since there is a normal distribution for the MFI values. Mean and standard deviation are now shown here and the text has been adapted accordingly. We apologize for the mistake of using a Mann-Whitney test despite the normal distribution in the original version of the manuscript.
- Line 128: The conclusion of sub-section 2.3 is too strong for the data presented. To support the conclusion that depletion of neutrophils leads to increased T cell activation, the experiment needs to be repeated with more activation markers assessed in vivo and/or ex vivo analysis of isolated T cells in context of proliferation/cytokine production. Otherwise, the conclusion of this section needs to be tempered.
We thank you for this comment and have revised our text accordingly.
- Line 191: the word “applying” doesn’t seem appropriate. Please rephrase.
- Line 194: It may provide more context to the readers if the authors include the fact that FcRn-mediated IgG transfer does take place between fetus and placenta in humans, similar to mice, but this doesn’t happen post-partum in human gut as it does in mice.
Thank you for these comments. We addressed the issues in the revised manuscript.
- Line 284: Please provide a justification for using a different route of administration for the isotype control.
Here, unfortunately, we are not clear what is meant by this comment. Depleting antibody and isotype control antibody were applied in exactly the same way. If this is incomprehensibly formulated at any point, we will be happy to change this, but we need more precise information on this again.
- Line 306: Please also provide a supplemental figure depicting the gating scheme used.
Thank you for this recommendation. We now added the gating strategies for peripheral T-cells and thymocytes as supplementary figure 4 to the revised version of our manuscript.
- Please add a summary or concluding statement at the end of each result sub-section.
Thanks a lot for this hint. We have included this in the revised version of our manuscript.

Reviewer 2 Report
This work describes antibody-mediated neutrophil depletion in newborn mice, and to show that this treatment modulates T-cell properties in the thymus and periphery.
1. Authors show the depletion of neonatal neutrophils in spleen, liver, and lung. It would be also important to show the effect in thymus, as the paper mostly focuses on thymic maturation.
2. Can you comment on the overall effect of Ly6G depletion on T cells? E.g. Why lung CD8 T cells were increased at P7 and reduced at P14 with Ly6G depletion?
3. Fig.4.F. In each column the DP/DN population shows two clusters. Why is the clustering, can you explain? Are these data from different time or different litters?
Author Response
- Authors show the depletion of neonatal neutrophils in spleen, liver, and lung. It would be also important to show the effect in thymus, as the paper mostly focuses on thymic maturation.
Thank you for this valuable comment. Since we did not initially expect neutrophilic cells in the thymus, unfortunately we only have data on the effect of the anti-Ly6G antibody on neutrophilic cells in the thymus at P14. Here we can see that there are very few neutrophils in the thymus at all and that they are completely depleted by the antibody. We added this information to the revised manuscript on page as Supplementary Figure 3.
- Can you comment on the overall effect of Ly6G depletion on T cells? E.g. Why lung CD8 T cells were in-creased at P7 and reduced at P14 with Ly6G depletion?
Thank you for this question. As already mentioned in the discussion part of our manuscript (page 8, last paragraph) the cause for the reversal effect on lung CD8 T-cells is not clear. We now added a new part to the discussion of our manuscript addressing this point in more detail (page 9, last paragraph).
- Fig.4.F. In each column the DP/DN population shows two clusters. Why is the clustering, can you explain? Are these data from different time or different litters?
Thank you very much for this question. Inspired by your comment we checked the data again and found that the clustering can actually be explained to a large extent by different litters. Due to the short time available for the revision, we were not able to examine further litters and therefore we have included this point as a limitation in the discussion section of the revised manuscript on page 9, last paragraph.
